

# Demistify: an LES and SCM intercomparison of radiation fog

Ian Boutle[1], Wayne Angevine[2], Jian-Wen Bao[3], Thierry Bergot[4], Ritthik Bhattacharya[5], Andreas Bott[6], Leo Ducongé[4], Richard Forbes[7], Tobias Goecke[8], Evelyn Grell[9], Adrian Hill[1], Adele Igel[10], Innocent Kudzotsa[11], Christine Lac[4], Bjorn Maronga[12], Sami Romakkaniemi[11], Juerg Schmidli[5], Johannes Schwenkel[12], Gert-Jan Steeneveld[13], and Benoît Vié[4]

[1]Met Office, Exeter, UK
[2]CIRES, University of Colorado, and NOAA Chemical Sciences Laboratory, Boulder, USA
[3]NOAA Physical Sciences Laboratory, Boulder, USA
[4]CNRM, Université de Toulouse, Météo-France, CNRS, Toulouse, France
[5]University of Frankfurt, Germany
[6]University of Bonn, Germany
[7]ECMWF, Reading, UK
[8]DWD, Offenbach, Germany
[9]CIRES, University of Colorado, and NOAA Physical Sciences Laboratory, Boulder, USA
[10]UC Davis, USA
[11]FMI, Kuopio, Finland
[12]University of Hannover, Germany
[13]Wageningen University, Netherlands

**Correspondence:** I. A. Boutle (ian.boutle@metoffice.gov.uk)

**Abstract.** An intercomparison between 10 single-column (SCM) and 5 large-eddy simulation (LES) models is presented for a radiation fog case study inspired by the LANFEX field campaign. 7 of the SCMs represent single-column equivalents of operational numerical weather prediction (NWP) models, whilst 3 are research-grade SCMs designed for fog simulation, and the LES are designed to reproduce in the best manner currently possible the underlying physical processes governing fog formation. The LES model results are of variable quality, and do not provide a consistent baseline against which to compare the NWP models, particularly under high aerosol or cloud droplet number (CDNC) conditions. The main SCM bias appears to be toward over-development of fog, i.e. fog which is too thick, although the inter-model variability is large. In reality there is a subtle balance between water lost to the surface and water condensed into fog, and the ability of a model to accurately simulate this process strongly determines the quality of its forecast. Some NWP-SCMs do not represent fundamental components of this process (e.g. cloud droplet sedimentation) and therefore are naturally hampered in their ability to deliver accurate simulations. Finally, we show that modelled fog development is as sensitive to the shape of the cloud droplet size distribution, a rarely studied or modified part of the microphysical parametrization, as it is to the underlying aerosol or CDNC.

## 1 Introduction

Most operational numerical weather prediction (NWP) centres will list errors in fog forecasting amongst their top model problems, with the requirement for improvement considered high-priority (Hewson, 2019). The key customer driving this is


the aviation sector, with ≈40% of all delays (≈50% of weather related delays) at busy airports (such as London Heathrow, Paris CDG, San Francisco and New Delhi) being due to low visibility events. In the best case, these delays are inconvenient for passengers and expensive for airline operators (Cook and Tanner, 2015; Kulkarni et al., 2019). However, in the worst case, fog can also be a significant danger and is the second most likely cause of weather related accidents (Gultepe et al., 2019; Leung
et al., 2020).

Despite this importance, there is no international community working together on improving fog modelling. The Global Atmospheric System Studies (GASS) panel facilitates projects which draw together researchers from around the globe to work on specific and targeted process studies. Utilising large-eddy simulation (LES) and single-column (SCM) versions of NWP models, previous projects (including under GABLS and GCSS) have made significant advances in the understanding and
modelling of stable boundary layers (Beare et al., 2006; Cuxart et al., 2006), turbulent clouds (van der Dussen et al., 2013; Neggers et al., 2017) and aerosol-cloud interactions (Hill et al., 2015). A new GASS project related to fog modelling therefore presents an opportunity to form a community and address the challenges together, building on the previous understanding of the multitude of processes at play in radiation fog.

A previous intercomparison of radiation fog in SCM models (Bergot et al., 2007) demonstrated that even before fog onset
there were considerable differences between models, and found the model skill to be low. This intercomparison considers a new generation of NWP-SCM models, with more complex physical parametrizations, and for the first time will compare LES models for the same radiation fog event. The key questions to be considered include:

– How well can models simulate the development of radiation fog?

– What are the key processes governing the development of radiation fog, i.e. aerosol, cloud microphysics, radiation,
turbulence, dew deposition, something else?

– Which of these processes are mostly responsible for the biases seen in current NWP models?

– What level of complexity is required from NWP models to adequately simulate these processes?

The initial phase of work, documented in this paper, will constrain the surface properties and focus primarily on the atmospheric development of fog. This will document the current state of LES and NWP fog modelling within the community, and
provide guidance on opportunities for improvements applicable to many models. Further stages of the project will then consider feedbacks through the land surface, more complicated cases with non-local forcing, and the representation of fog in climate models, something which has rarely been looked at in the literature.

## 2  Intercomparison design and participants

The first intensive observational period (IOP1) of the Local and Non-local Fog Experiment (LANFEX, Price et al., 2018)
presented a relatively simple case of fog forming in a nocturnal stable boundary layer, developing over several hours into turbulent, optically-thick fog. However, NWP modelling of this event (Boutle et al., 2018) showed significant errors in the

structure and evolution of the fog. Therefore we base the intercomparison around a slightly idealised version of IOP1. The case is based at the Met Office observational site at Cardington, UK (52.1015N, 0.4159W) and occurred on the night of 24-25 November 2014. Models are initialised from the 17 UTC radiosonde profile and forced throughout the night by the observed

surface temperature. No other forcing is used, to keep the case simple and allow for maximum participation amongst modelling centres. Forcing with surface temperature also constrains the problem to an atmospheric one, focussing on the cloud, radiation and turbulence interaction. In reality, patchy fog began to form around 18 UTC, with persistent fog and visibilities around 100 m from 20 UTC for 12 hours before clearance.

Because of the sensitivity to cloud and aerosol processes previously discussed in Boutle et al. (2018), we request two

simulations from all participants. For models which do not represent aerosol processing, the cloud droplet number concentration (CDNC) should be prescribed (if possible) as:

- c10: fixed cloud droplet number concentration of 10 cm$^{-3}$,

- c50: fixed cloud droplet number concentration of 50 cm$^{-3}$,

For models which do represent aerosol processing, the accumulation mode aerosol should be prescribed as:

- a100: initial accumulation mode (0.15 $\mu$m diameter, $\sigma$=2) aerosol of 100 cm$^{-3}$,

- a650: initial accumulation mode (0.15 $\mu$m diameter, $\sigma$=2) aerosol of 650 cm$^{-3}$.

Experiments c10 and a100 will be referred to as "low" aerosol/CDNC simulations, whilst c50 and a650 will be referred to as "high" aerosol/CDNC simulations. The aerosol setup is complicated slightly, as some of the more sophisticated aerosol processing models also require specification of the Aitken and coarse mode aerosols, which are prescribed (as in Boutle et al.,

2018) as 1000 cm$^{-3}$ with mean diameter 0.05 $\mu$m and 2 cm$^{-3}$ with mean diameter 1 $\mu$m. Vié et al. (2021) discuss how it is only really sensible to impose these additional aerosols in models which represent prognostic supersaturation of liquid water, otherwise excessive activation of the Aitken mode aerosol into cloud droplets occurs.

Although the surface temperature is specified, many models still require some parametrization of the surface characteristics (to estimate the turbulent fluxes into the atmosphere), which is set as a flat, homogeneous, grass surface with the following

parameters:

- Momentum roughness length ($z_{0m}$): 0.1 m,

- Heat roughness length ($z_{0h}$): 0.001 m,

- Leaf area index: 2,

- Albedo: 0.25,

- Emissivity: 0.98.





| Institution | Model | Type | Experiments submitted | Lead participant | Reference |
|---|---|---|---|---|---|
| Bonn University | MiFog | SCM | a100, a650 | Andreas Bott | Bott et al. (1990) |
| Bonn University | PaFog | SCM | a100, a650 | Andreas Bott | Bott and Trautmann (2002) |
| CIRES/NOAA | WRF | SCM* | c10, c50, a100, a650 | Wayne Angevine | Angevine et al. (2018) |
| UC Davis | RAMS | LES | c10, c50, a100, a650 | Adele Igel | Cotton et al. (2003) |
| DWD | ICON | SCM* | c10, c50 | Tobias Goecke | Bašták Ďurán et al. (2021) |
| ECMWF | IFS | SCM* | c10, c50 | Richard Forbes | |
| FMI | UCLA-SALSA | LES | a100, a650 | Innocent Kudzotsa | Tonttila et al. (2017) |
| Frankfurt University | COSMO | SCM* | c10, c50 | Ritthik Bhattacharya | Baldauf et al. (2011) |
| Hannover University | PALM | LES | a100, a650 | Johannes Schwenkel | Maronga et al. (2020) |
| Met Office | Unified Model | SCM* | c10, c50 | Ian Boutle | Bush et al. (2020) |
| Met Office | MONC | LES | c10, c50, a100, a650 | Adrian Hill | Dearden et al. (2018) |
| Meteo France | Meso-NH | SCM* | c10, c50, a100, a650 | Leo Ducongé | Lac et al. (2018) |
| Meteo France | Meso-NH | LES | c10, c50, a100, a650 | Leo Ducongé | Lac et al. (2018) |
| NOAA | FV3-GFS | SCM* | c300 | Evelyn Grell | Firl et al. (2020) |
| Wageningen University | d91 | SCM | c10, c50 | Gert-Jan Steeneveld | Duynkerke (1991) |

**Table 1.** Modelling centres, lead participants, models and model simulations submitted. * denotes the SCMs that have the physics package and vertical resolution of operational NWP models.

| Model | Grid-length (dx, dz) | Aerosol processing | Microphysics type | Prognostic supersaturation | Cloud droplet settling | Subgrid turbulence |
|---|---|---|---|---|---|---|
| RAMS | 4 m, 1.5 m | Accumulation | Bulk | N | Y | TKE |
| UCLA-SALSA | 4 m, 1.5 m | Full | Bin | Y | Y | Smagorinsky |
| PALM | 1.5 m, 1.5 m | Accumulation | Bulk | N | Y | TKE |
| MONC | 4 m, 1.5 m | Accumulation | Bulk | N | Y | Smagorinsky |
| Meso-NH | 4 m, 1.5 m | Accumulation | Bulk | N | Y | TKE |

**Table 2.** LES model details: horizontal (dx) and vertical (dz) grid-length, type of aerosol processing, microphysics parametrization details and type of subgrid turbulence scheme (TKE=Turbulent kinetic energy closure).





| Model | Grid-length (lowest level, levels below 150 m) | Aerosol processing | Microphysics type | Prognostic supersaturation | Cloud droplet settling | Subgrid turbulence |
|---|---|---|---|---|---|---|
| MiFog | 0.5 m, 61 | Full | Bin | Y | Y | TKE |
| PaFog | 0.5 m, 61 | Full | Bulk | Y | Y | TKE |
| WRF | 12 m, 6 | Accumulation | Bulk | N | Y | TKE+EDMF |
| ICON | 10 m, 3 | None | Bulk | N | Y | TKE |
| IFS | 10 m, 6 | None | Bulk | N | N | EDMF |
| COSMO | 10 m, 7 | None | Bulk | N | N | TKE |
| Unified Model | 2.5 m, 6 | None | Bulk | N | Y | K1+NL |
| Meso-NH | 5 m, 7 | Accumulation | Bulk | N | Y | TKE |
| FV3-GFS | 21 m, 3 | None | Bulk | N | N | EDMF |
| d91 | 3.3 m, 27 | None | Bulk | N | Y | K1 |

**Table 3.** SCM model details: height of lowest model level and number of levels below 150 m, type of aerosol processing, microphysics parametrization details and type of subgrid turbulence scheme (EDMF=Eddy-diffusivity mass-flux closure, K1=local first order closure, NL=non-local/counter-gradient transport).

Evapotranspiration should be unrestricted, to avoid complexities associated with soil moisture and land-surface models, although in practice the fluxes are into the surface for most of the night and so this simplification is of limited importance.

Table 1 shows the model configurations that have been submitted and are analysed in this paper, whilst Tables 2 and 3 give some further relevant details about the setups of the LES and SCM models respectively.

## 3 Results

### 3.1 Liquid water path evolution

Figure 1 presents an initial view of the submitted models, separated by their class (LES or SCM) and aerosol or CDNC (low or high). The first thing to note is that all models do at least form fog, but beyond this there is very little consistency between models.

The observations are most consistent with the low aerosol/CDNC setup. For the SCM runs, only MiFog, Meso-NH, UM and d91 have liquid water path (LWP) evolution in line with the observations, although PaFog, IFS and WRF are reasonably close. The other models considerably over-estimate the LWP. In general, the LES runs are in closer agreement with each other and the observations, but considerable spread exists between them for the high aerosol/CDNC runs. With the exception of ICON and FV3 (which does not represent variable CDNC), all models show substantial variation between the low and high aerosol/CDNC setups, producing higher LWP with greater aerosol/CDNC.



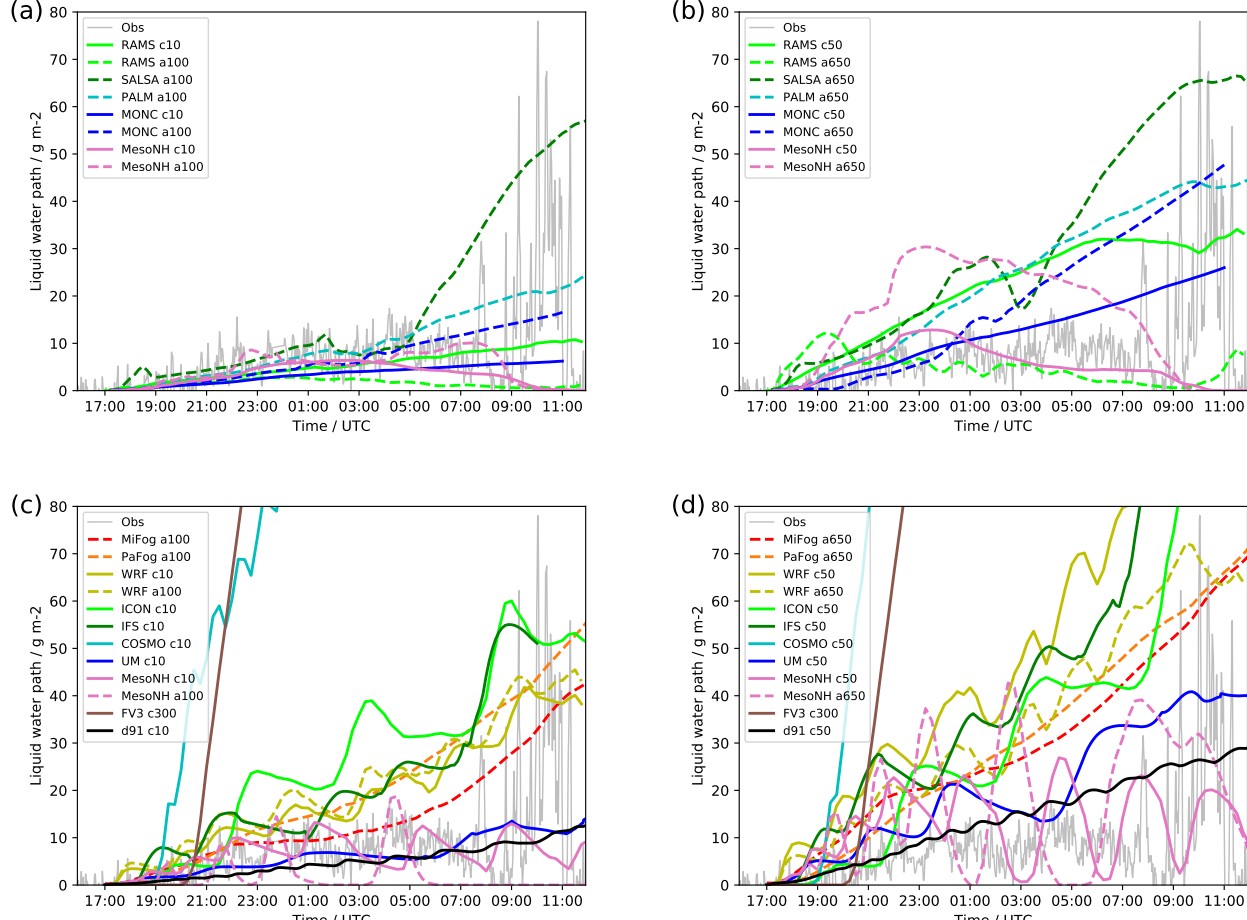

**Figure 1.** Liquid water path observed and modelled by (a) low aerosol/CDNC LES, (b) high aerosol/CDNC LES, (c) low aerosol/CDNC SCMs and (d) high aerosol/CDNC SCMs.





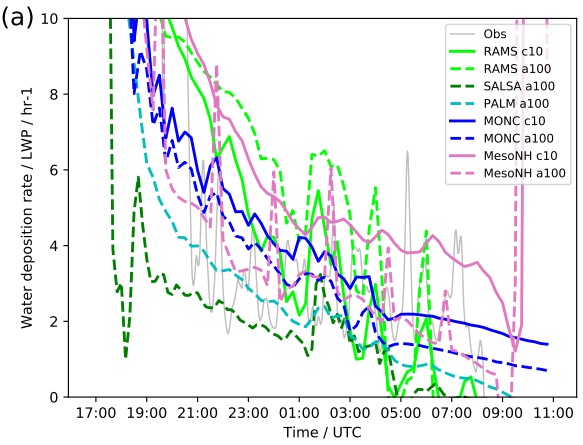
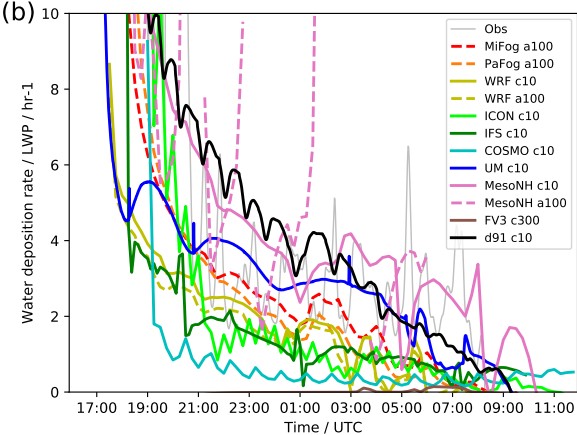

**Figure 2.** Water deposition rate divided by liquid water path observed and modelled by (a) low aerosol/CDNC LES, (b) low aerosol/CDNC SCMs.

To first order, the most important factor in determining the LWP evolution of all models is the rate at which water is deposited from the atmosphere to the surface. The observations (see Boutle et al., 2018, Fig 4a) are broadly constant at around $20\,\mathrm{gm^{-2}hr^{-1}}$ throughout the night, and most models achieve this value despite the wildly varying LWP (possibly because the water deposition is constrained by the long-wave cooling of the atmosphere). Because the water deposition rate is strongly affected by the LWP, we must therefore normalise it before comparing the models, which is shown in Figure 2. This shows a clear link between the deposition rate and LWP – models which do not deposit enough water onto the surface end up with LWP values which are too high, and models which deposit too much water onto the surface end up with LWP values which are too low.

The reasons for the varying water deposition rate are very model dependent, although we can try to summarise some consistent themes in the SCMs:

– Models which do not represent cloud droplet sedimentation – these models (FV3, COSMO, IFS) are significantly hampered by their lack of this process, which is likely to be the dominant mechanism of water removal in reality. IFS is able to compensate to a certain extent by autoconverting significant amounts of fog into precipitation and removing it that way, which explains its lower LWP than COSMO or FV3 which are unable to do this. Improvements here should be easy to achieve via modifications to the microphysical parametrization.

– Models which produce excessive positive surface latent heat flux (Fig. 5) – these models (WRF, COSMO) will always struggle to deposit enough water through microphysical processes because it is being constantly replenished via evaporation from the surface. Understanding the mechanisms behind this error can be tricky, as it may not simply be an issue with the turbulent exchange parametrizations, but could also be a feedback. For example, as discussed in Boutle et al.





(2018), forming fog which is slightly too optically thick can drive an erroneous positive flux, which in turn leads to further development of thicker fog.

- The precise nature of the microphysical parametrizations responsible for water deposition – even models which represent all processes and maintain a low latent heat flux (ICON, UM, Meso-NH) can have large discrepancies because of how the different water deposition rates feed back onto model evolution. This suggests that more work is required on the basic
observations, understanding and modelling of water deposition. For example Meso-NH is the only model to represent turbulent deposition of droplets in addition to sedimentation, giving it one of the highest deposition rates.

The LES models may be closer in their behaviour, but still show some similar traits to the SCMs. In particular, the models with the highest deposition rates tend to have the lowest LWP, and visa-versa. However, the mechanism by which this is achieved can be considerably different between the models. RAMS-c10 for example has a significant positive latent heat flux, which is
balanced by a larger cloud droplet sedimentation rate than any other LES, to give an overall water deposition rate and LWP comparable to the other models. Differences like this show why it is difficult to use the LES as process models, because although they are producing more consistent behaviour, the processes by which they achieve it are not consistent.

The one LES (and indeed SCM) model which doesn't appear to follow the pattern is Meso-NH-c10, which has one of the highest water deposition rates of any of the models, yet manages to achieve a reasonable fog simulation in all cases. This arises
because it simulates a very low effective radius (Fig 3), resulting in very strong absorption and emission from the fog layer, helping the fog to grow despite the high water deposition. The reason for the low effective radius appears to be the use of the Martin et al. (1994) parametrization with a default 'land' setup, i.e. it is using a high ($300 \, \mathrm{cm^{-3}}$) assumed CDNC value in the effective radius parametrization, rather than the actual CDNC used by the microphysical parametrization. The Meso-NH-a100 simulation, which has a consistent link between cloud droplet number and effective radius, shows a response more consistent
with the other models. This highlights the importance of using consistent assumptions between radiation and microphysical parametrizations.

The RAMS-a100 simulation has almost the opposite effect, with a high effective radius resulting in a very low LWP. This however arises because the model rapidly depletes all of the aerosol in the atmosphere, and therefore has nothing to activate into cloud droplets. As a consequence, after the initial fog formation, no new small droplets are formed, but the droplets which do
exist grow in size and sediment out, resulting in a very low liquid water path. This is particularly noticeable in the RAMS-a650 simulation, which has the lowest LWP of any model in the 'high' experiment. Figure 4(b) shows that this is linked to a very low CDNC, despite the high initial aerosol concentration, because most of the aerosol has been depleted. Fig. 4(b) also shows an interesting clustering between the full aerosol processing models, which predict CDNC values in the range 40-60 $\mathrm{cm^{-3}}$, and the accumulation only models which predict CDNC values in the range 70-90 $\mathrm{cm^{-3}}$. This shows that even though the latter
group are only considering a subset of the full aerosol distribution, they may still be over-estimating the activation occurring in the fog layer. However, Fig. 1(b and d) shows that this clustering in the CDNC value does not equate to a clustering in the LWP evolution, demonstrating that there are larger differences between the models than the predicted CDNC value.

**Figure 3.** Effective radius observed and simulated by the low aerosol/CDNC (a) LES, and (b) SCMs, at 00 UTC.





**Figure 4.** CDNC observed and from the aerosol processing models at 00 UTC, for (a) low aerosol, and (b) high aerosol.





It is worth briefly discussing the oscillations in LWP seen in the SCM models. This is a known feature of fog SCM simulations, and has been discussed previously by Tardif (2007). Long-wave (LW) cooling from the fog top is the key driver of the

fog layer deepening. However, with the coarse vertical grid of the SCM models, the fog can only deepen in discrete units, when the top grows by a single model level. The LW cooling therefore causes the LWP to erode, until such time as the fog can jump up a level, leading to a large increase in LWP. Hence the oscillations are created. All of the SCMs with coarse vertical grids show some oscillations, although the severity of them differs significantly. By far the simulation to suffer most is Meso-NH-a, which appears to have a further complicating feedback from the microphysics. When the fog top jumps up a level, the increase

in LWP triggers significant precipitation formation, which quickly removes a large amount of water from the atmosphere. This microphysical feedback does not disappear when running Meso-NH-a at higher vertical resolution, whereas the oscillations in Meso-NH-c do (not shown).

### 3.2    Surface fluxes and boundary layer structure

A key feature of this fog event, and indeed many fog events, is the slow transition from a stable boundary layer with optically

thin fog to a well-mixed boundary layer with optically thick fog. How this transition evolves is of key interest from a forecasting perspective, as it will determine the depth and intensity of the fog layer, and ultimately its duration into the following morning.

Interestingly, the LES models show greater variability in the surface heat flux (Fig. 5a) than they did for the liquid water path. Whilst there is some hint towards the expected trend that models which are optically thickest (PALM, Meso-NH-c10) will generate a positive surface heat flux and well-mixed fog layer first, RAMS-c10 sits as a clear outlier here generating the

strongest positive surface heat flux whilst having one of the thinnest (optically and physically) fog layers. It achieves this by forming a shallow, but well-mixed layer in which the fog exists (Fig. 6a), capped by a strong inversion. RAMS does indeed have a higher downwelling LW radiation, which would promote development of a well-mixed fog layer. However, why it keeps shallow and does not feed back like it does in Meso-NH is interesting, suggesting lower entrainment across the inversion. The result is that RAMS has the lowest fog top of all the LES models (Fig. 7a).

The SCMs show a similar trend to the LES models, with many producing a positive surface heat flux and well-mixed boundary layer structure. But as always, there are interesting outliers. WRF and IFS, which both slightly over-estimate the LWP (and thus create fog which is optically and physically too thick) manage to maintain a stable potential temperature profile throughout the fog layer (Fig. 6b). IFS is particularly interesting as it manages to do this with a positive sensible heat flux. This suggests that their turbulence parametrizations are able to adapt to the local environmental conditions better than, e.g. the UM,

which as discussed in Boutle et al. (2018) forms a well-mixed boundary layer almost instantly after a positive surface heat flux is diagnosed. It's also worth discussing FV3, which is the only model which produces a negative heat flux. This is possibly due to its poor vertical resolution, with the lowest model level being approximately double the height of any other model, meaning the lowest level temperature is very warm relative to the surface. In its default setup, FV3 also produced a very negative latent heat flux, which prevented any fog formation and needed to be restricted to enable fog to form.





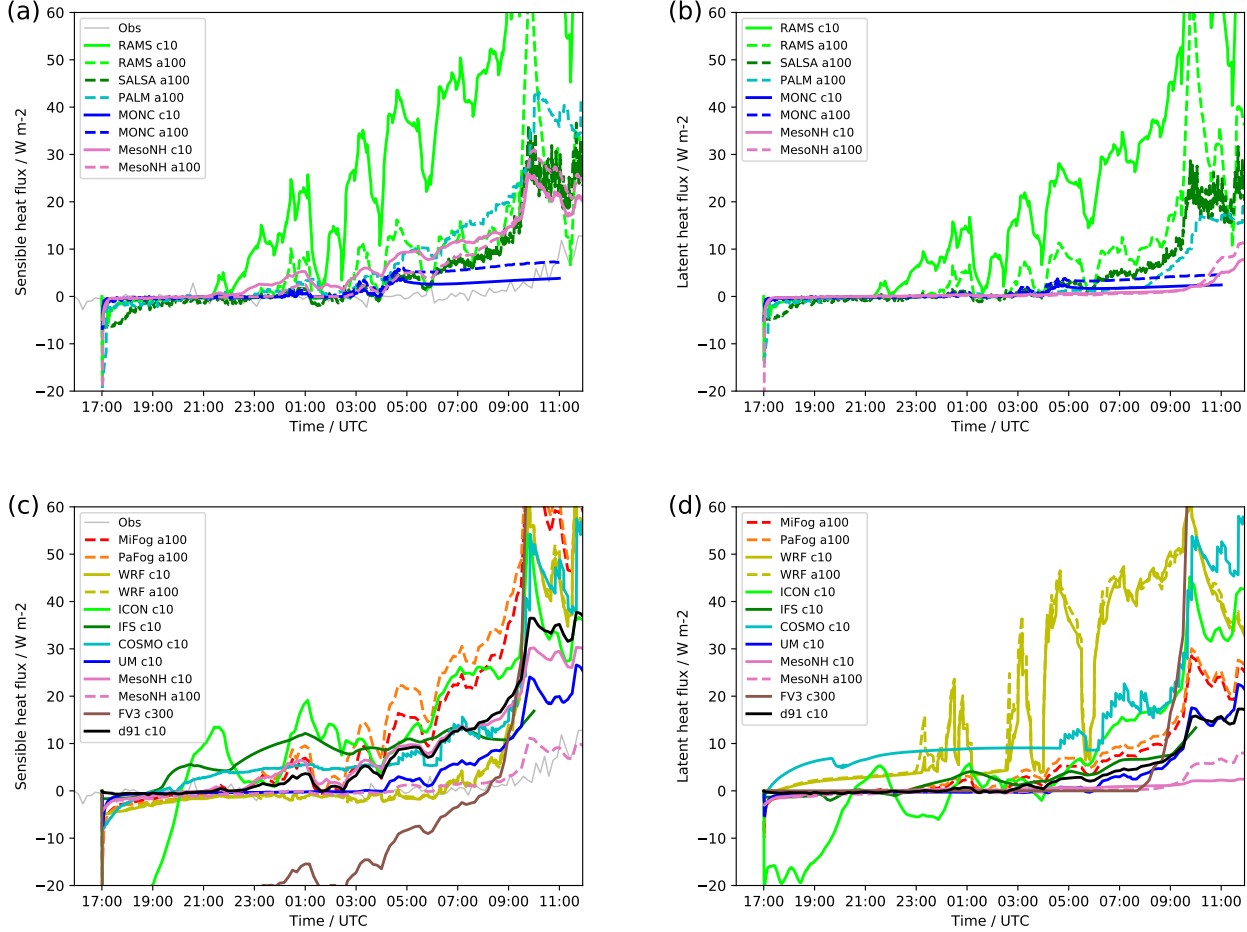

**Figure 5.** Sensible (left) and latent (right) heat flux observed and modelled by (top) low aerosol/CDNC LES, (bottom) low aerosol/CDNC SCMs.

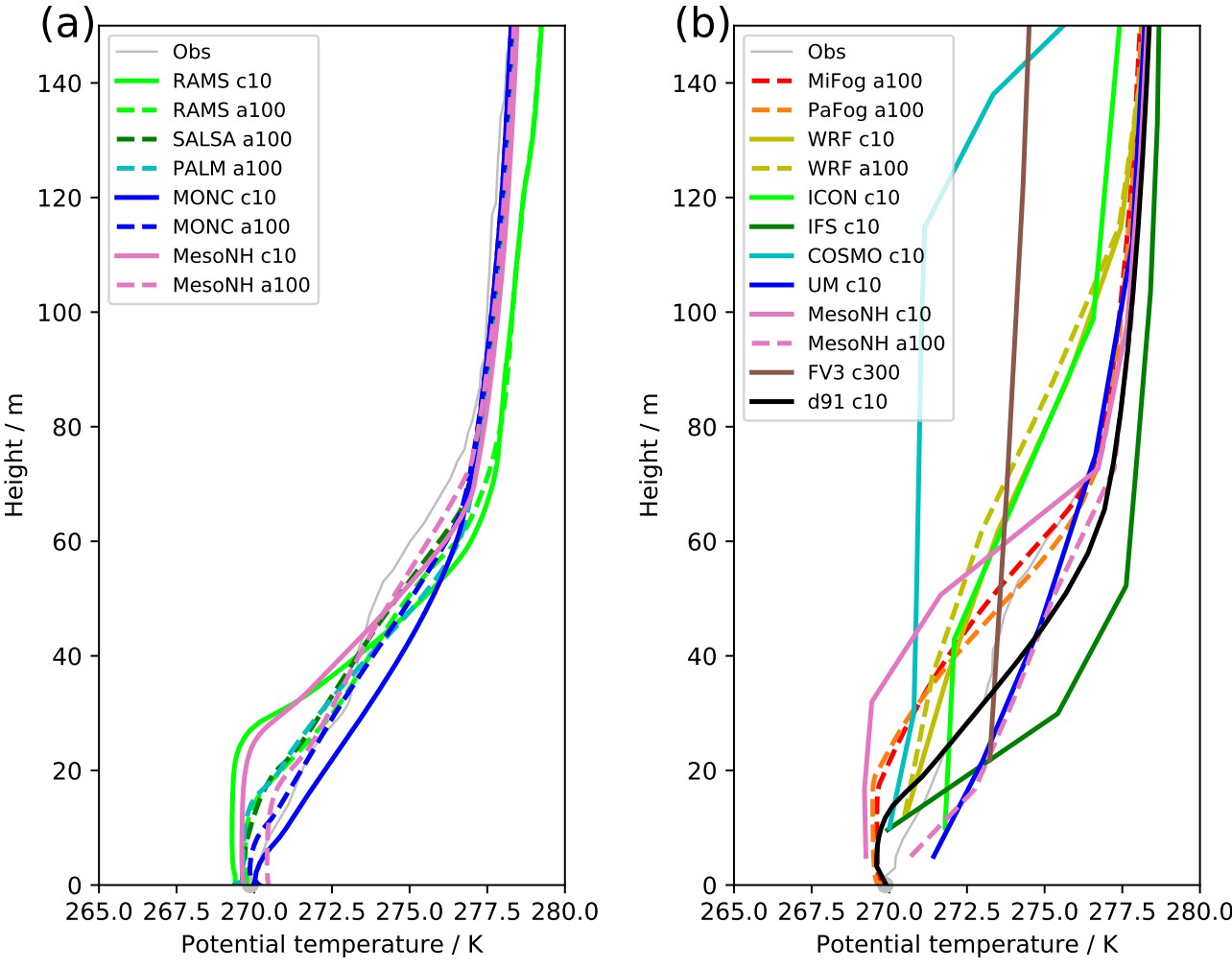

**Figure 6.** Potential temperature observed and simulated by (a) the low aerosol/CDNC LES models, and (b) the low aerosol/CDNC SCMs at 00 UTC.



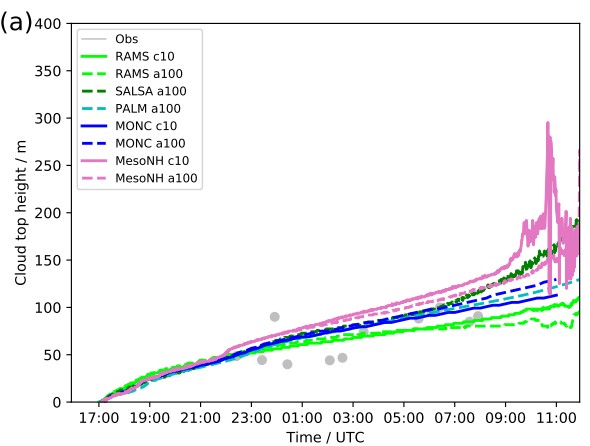 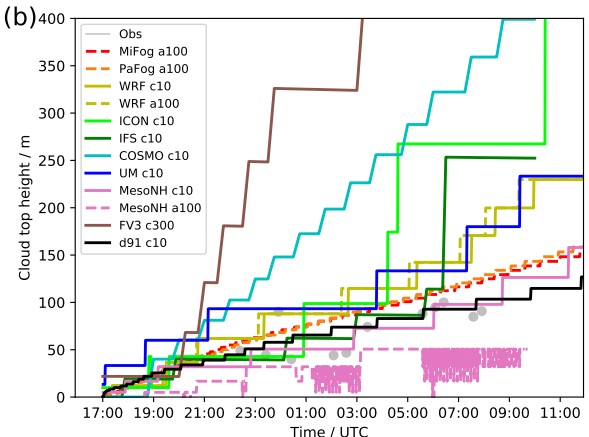

**Figure 7.** Fog top height observed and modelled by (a) low aerosol/CDNC LES, (b) low aerosol/CDNC SCMs.

## 3.3 Forecasting considerations

In terms of fog impact, particularly to the aviation sector, correctly modelling fog clearance after sunrise is key to forecasting airfield clearance time and allowing full take off/landing rates to resume. There are a number of aspects of the intercomparison which complicate the simulation of the morning transition – the unrestricted evaporation is unrealistic for a true land-surface; the observed surface temperature warming is representative of fog which has dissipated in reality for a number of reasons not simulated by the LES and SCMs (particularly overlying cloud cover, which is responsible for the observed increase in LWP after 08 UTC). However, comparison between how the models deal with this situation can still provide some useful insights. As shown in Figure 1, MesoNH-LES is the only model which completely dissipates the fog during the morning. Most models' fog evolution seems broadly unaffected by the increasing surface temperature and short-wave radiation, except for SALSA, where it drives a large increase in LWP. There are essentially two competing mechanisms at work here. The increase in surface temperature will drive a strong positive surface moisture flux, promoting fog development. However, direct short-wave heating of the fog layer and heating due to the rise in surface temperature and positive surface heat flux will counteract this. The consequences for fog development are therefore model dependent, based on the relative importance of these processes.

If the surface temperature was not prescribed, the key quantity driving dissipation would be the downwelling short-wave radiation (as this would drive the surface heating), which is shown in Figure 8. The figure shows that the degree of variation between models is large (over 250 $\mathrm{Wm}^{-2}$), with similar uncertainty between the LES and SCM models. To first order, the key reason for differences in the downwelling short-wave is the LWP at sunrise – the models with the highest LWP have the lowest downwelling SW and vice-versa. Optical properties of the fog appear to be much less significant here – for example comparing the UM and Meso-NH-c10 SCM simulations – Meso-NH-c10 only has a slightly smaller LWP, but coupled with its much smaller effective radius results in almost identical downwelling SW evolution. What is clear is that there is a huge range





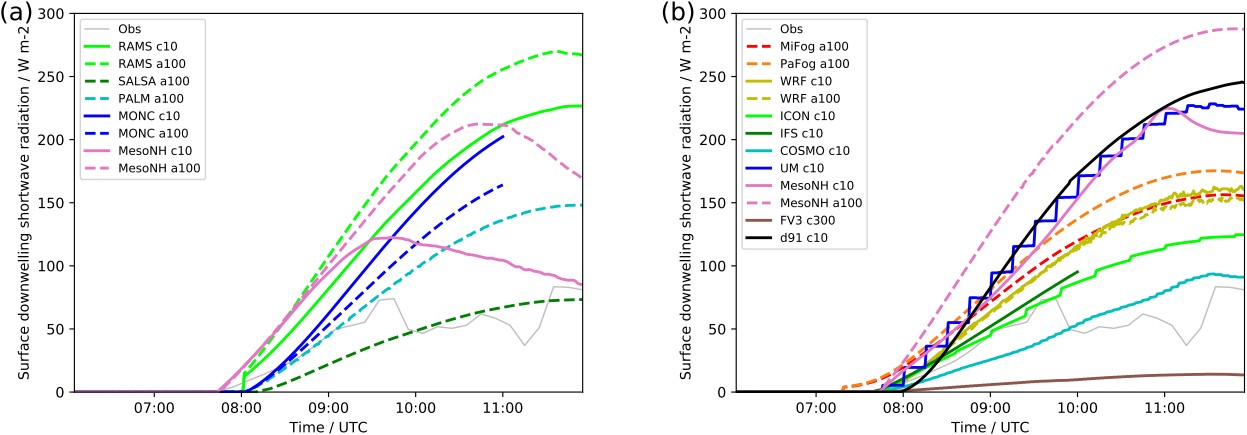

**Figure 8.** Surface downwelling short-wave radiation observed and modelled by (a) low aerosol/CDNC LES, (b) low aerosol/CDNC SCMs.

in potential fog evolution and dissipation times driven by differences in the fog development during the night-time. Having knowledge of how realistic a model forecast of fog development through the night-time is (e.g. via real time observations) may enable a forecaster to understand how reliable the forecast for morning dissipation is.

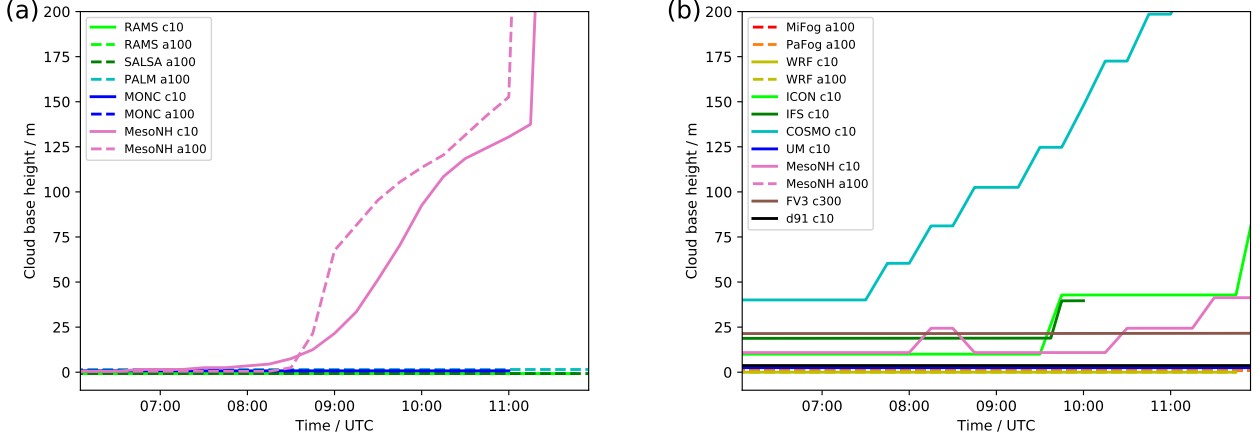

**Figure 9.** Cloud base height modelled by (a) low aerosol/CDNC LES, (b) low aerosol/CDNC SCMs.

Another forecasting consideration is whether the fog will indeed dissipate, or whether it will lift into low stratus. Figure 9 shows the cloud base height ($q_c > 0.01$ gkg$^{-1}$) during the morning period for the LES and SCM models, demonstrating that there is significant variety in model simulation of this behaviour. Whilst most models keep the fog firmly on or near the ground, Meso-NH LES and COSMO SCM lift the cloud base significantly, with cloud base height exceeding 60 m (the





threshold typically usually used by aviation for instigating low visibility procedures) by 8 UTC – 9 UTC. The difference here (and elsewhere) between Meso-NH LES and SCM is of particular interest, because the physics package of both models is identical, meaning that differences must arise because of the lower vertical resolution in SCM, or because the 1D parametrized

turbulence in the SCM is acting differently to the 3D resolved turbulence in the LES.

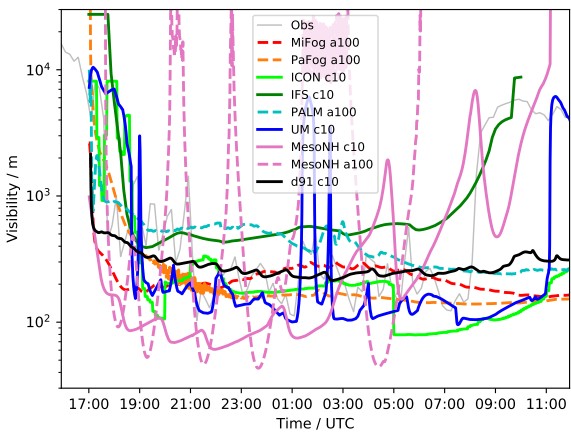

**Figure 10.** Screen-level visibility observed and predicted by all models including a specific visibility parametrization. The model parametrizations are all different – some use simple empirical functions, whilst others calculate the extinction coefficient from the droplet size distribution.

    Finally, we discuss some of the typical metrics used by decision makers when forecasting fog events. Figure 10 shows the screen level visibility as predicted by all models incorporating a visibility parametrization. Given the differences seen elsewhere in the fog evolution, the level of agreement between models here is somewhat surprising. Most models are forecasting visibility in the 100-300 m range for most of the night, in line with observations. IFS and PALM are forecasting slightly larger

visibilities ($\approx$500 m), but still below the thresholds typically used by aviation decision makers (600 m), whilst only Meso-NH produces visibilities below 100 m. Most models also retain low visibilities well into the morning period, with only Meso-NH, IFS and eventually the UM forecasting a clearance in this metric. The consistent behaviour may, in part, be due to the tight linkage between screen-level and surface variables in many models, as with the surface temperature prescribed, the screen-level temperature does not deviate far from the observations (not shown). However, it also raises caution against the use and

interpretation of such variables, if they can seemingly produce such similar results despite such obvious differences in the actual simulation of fog within the models. To truly understand and interpret an NWP fog forecast requires much more than simply looking at the predicted visibility, especially in more marginal cases than this one.

    Table 4 shows for all models the onset and dissipation time of the fog event, and the maximum height reached by the fog layer. This summarises many of the themes discussed so far in the paper. The initiation of fog is handled well by all models,

with the initiation happening between 17 and 18 UTC in all but 2 of the models. Many models show that low visibility (LVP)





| | Fog onset | LVP start | Fog dissipation | LVP end | Max fog top (m) |
|---|---|---|---|---|---|
| **Observations** | | 17:45 | | 08:04 | ≈100** |
| MiFog a100 | 17:00 | 17:11 | >12 | >12 | 153 |
| PaFog a100 | 17:15 | 18:25 | >12 | >12 | 159 |
| WRF c10 | 17:30 | | >12 | | 230 |
| WRF a100 | 17:30 | | >12 | | 230 |
| RAMS c10 | 17:30 | | >12 | | 110 |
| RAMS a100 | 17:30 | | >12 | | 95 |
| ICON c10 | 17:30 | 18:49 | 11:45 | >12 | 503 |
| IFS c10 | 17:52 | 18:37 | >12 | 06:37 | 253 |
| SALSA a100 | 17:10 | | >12 | | 196 |
| COSMO c10 | 19:00 | | 08:00 | | 489 |
| PALM a100 | 17:00 | 19:09 | >12 | >12 | 130 |
| UM c10 | 17:01 | 18:41 | >12 | 11:08 | 233 |
| MONC c10 | 17:30 | | >12 | | 113 |
| MONC a100 | 17:30 | | >12 | | 129 |
| MesoNH SCM c10 | 17:15 | 17:15 | >12 | 09:19 | 158 |
| MesoNH SCM a100 | 17:45 | 17:45 | 06:00 | 05:30 | 51 |
| MesoNH LES c10 | 17:30 | | 09:15 | | 295 |
| MesoNH LES a100 | 17:30 | | 08:45 | | 271 |
| FV3 c300 | 20:15 | | >12 | | 622 |
| d91 c10 | 17:04 | 17:10 | >12 | >12 | 127 |
| **LES mean** | 17:23 | | 09:00* | | 167 |
| **LES range** | 0:30 | | 0:30* | | 200 |
| **SCM mean** | 17:44 | 17:59 | 08:35* | 08:08* | 267 |
| **SCM range** | 3:15 | 1:39 | 5:45* | 5:38* | 571 |

**Table 4.** Selected forecasting metrics for each model, as observed, and the mean and range of results for the LES and SCM models combined. Fog onset/dissipation is defined by liquid water below 60 m, whilst typical airfield low visibility procedures (LVP) are defined by visibility <600 m and cloud base <60 m. '>12' denotes models which did not dissipate fog by the end of the simulation. *Dissipation statistics are only calculated from the models which dissipated fog during the morning. **Recorded around 08 UTC just before the fog dissipated.





occurs some time after fog onset, demonstrating that the models are able to capture an initial period of thin fog where visibility remains good. The dissipation phase is much poorer, with most models persisting fog until the end of the simulation. Only a minority of models break the fog during the morning period, and with no consistency in how this is done – some lifting it into stratus whilst others clear it entirely. Whilst a few models do thin the fog sufficiently for LVP to end, it would clearly

be very difficult to provide guidance to customers based on this ensemble set. The mean fog depth simulated by the SCMs is approximately 100 m higher than that from the LES, and at the very top end of the LES range. This is symptomatic of the SCM behaviour in producing fog which is too thick, and is likely to lead to fog persisting for too long into the daytime.

## 4    Parametrization Sensitivity

To explore some of the themes and relationships shown in Section 3.1, in this section we focus on 2 SCMs (COSMO and UM)

and 1 LES (MONC), modifying several parametrizations to confirm the speculated reasons for fog differences. The first and most simple test, using the UM, is to switch off cloud droplet sedimentation entirely (similar to COSMO, FV3 or IFS). This is shown in Figure 11.

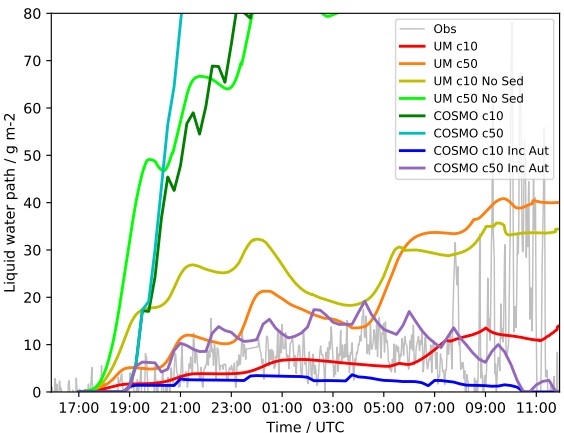

**Figure 11.** Liquid water path observed and modelled with low and high CDNC values, from the UM with or without cloud droplet sedimentation, and COSMO with low and high autoconversion rates (see caption).

The removal of cloud droplet sedimentation leads to large increases in the liquid water path, for both CDNC values. Clearly the presence or absence of cloud drop sedimentation is more important than the prescription of CDNC value. This also confirms

why models which do not represent this process produce a fog layer which is too thick.

Whilst implementing cloud droplet sedimentation in models which do not have it is ultimately the most physically realistic way of improving fog simulation, we can also investigate, using COSMO, how simulations might be improved with the parametrizations at hand. The autoconversion in COSMO (Seifert and Beheng, 2001) is proportional to the 4th power of cloud





water content, and therefore produces very little autoconversion at low water contents. Reducing the power (to 3.1) allows the
autoconversion rate to be increased at low water contents. As shown in Figure 11, the consequence of this is a much improved
fog simulation, again confirming that the rate of water loss from the atmosphere is the dominant mechanism governing the fog
LWP. This also shows why IFS, which uses Khairoutdinov and Kogan (2000) autoconversion (power 2.47) is able to produce
lower and more realistic LWP evolution without cloud droplet sedimentation. It is worth clarifying again that this is not a real-
istic model improvement we would suggest implementing – fog droplets are small and collision-coalescence is rare, therefore
autoconversion should not be happening.

For models which do simulate cloud droplet sedimentation, how sensitive is the fog development to the precise details of
the parametrization? This is explored with the MONC LES, by varying the shape parameter, $\mu$, used in the cloud droplet size
distribution:

$$N(D) = N_0 D^\mu e^{-\lambda D}, \tag{1}$$

where $N$ is the number of drops of diameter $D$, $N_0$ is the intercept parameter and $\lambda$ is the slope parameter. Miles et al. (2000)
have shown that $\mu$ in the range 2-5 is most commonly found in stratiform clouds, but values in the range 0-25 have been found
in observations. The default value used in MONC is $\mu = 2.5$, and Figure 12 shows a sensitivity study varying $\mu$ between 0 and
10.

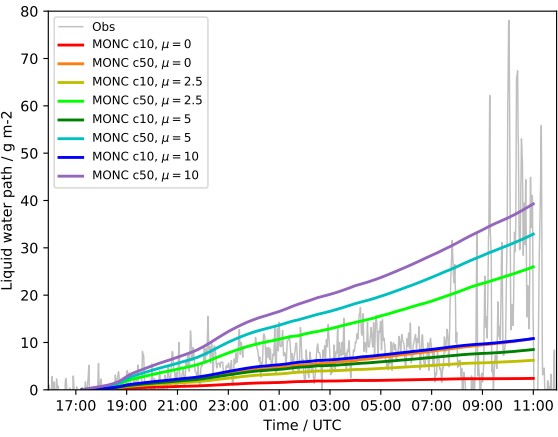

**Figure 12.** Liquid water path observed and modelled by MONC with low and high CDNC values, with varied values of the shape parameter
$\mu$ (see caption).

Once again, this relatively minor change to part of the microphysical parametrization has a larger effect on fog evolution than
the prescribed CDNC value, showing the importance of fundamental parametrization development. It's also interesting to note
that with the reduction of $\mu$, which increases droplet sedimentation rates, it is actually possible to produce a fog layer which
is too thin – no other model has shown this so far. This acts to highlight why even when all processes are represented within



a model, large differences in fog evolution can still be seen because the fog evolution is so sensitive to small parametrization changes.

This section has shown that even for a highly constrained scenario, the microphysics of fog remains a very uncertain process. We could, for example, recommend that future field campaigns focus on ascertaining with better accuracy the parameters of bulk microphysics parametrizations (for example $\mu$). However, existing observations show that frequently size distributions are bimodal in nature (Wendisch et al., 1998; Price, 2011), and therefore we should question whether microphysics parametrizations imposing a Gamma distribution are even the appropriate tool for fog simulation. Bin microphysics parametrizations (such as that employed in SALSA or MiFog) offer a better ability to simulate the evolution of the size distribution, and certainly these models are among the best performing in this intercomparison. Recently, Schwenkel and Maronga (2020) demonstrated the use of a Lagrangian cloud model (LCM) for fog simulation, and found (consistent with this work) that the LCM tended to produce greater sedimentation rates and lower liquid water paths than a bulk scheme, due to its evolution of the size distribution. However Bin schemes and LCMs are likely to be prohibitively expensive for operational implementation, and therefore how to best represent this behaviour in operational models remains an open question. They also contain many more degrees of freedom, and thus it is important that future observational campaigns focus not just on the mean value of microphysical parameters, but also the time and space variability of the full size distribution, to allow accurate evaluation of Bin schemes and LCMs.

## 5   Conclusions

If nothing else, this paper has highlighted why fog remains such a difficult forecasting challenge. The level of comparability between our most detailed process models – LES – is much lower than has been seen in previous intercomparison studies of other boundary-layer or cloud regimes (Beare et al., 2006; van der Dussen et al., 2013). This is largely due to the huge role microphysics plays in fog development, and uncertainties inherent in the representation of a process which is still entirely parametrized in LES. However, there were also strong differences seen in the surface fluxes and turbulent structure within the LES models. Whilst throughout the bulk of the fog layer the simulations were well enough resolved, near the surface the sub-filter scale flux clearly becomes dominant and provides an additional source of uncertainty not seen with higher-level clouds. This effectively means that LES cannot be considered an adequate baseline (or truth) against which to compare NWP models. Therefore our first recommendation must be for continued investment in observational understanding of real fog events, particularly to understand the high-frequency (in time and space) variability that exists in fog. This must be linked to continued development of LES models to a state where they can provide an adequate substitute for real observations.

285     For the SCMs, it is clear that improvements have been made since the previous intercomparison of Bergot et al. (2007), as a very good consistency between models in the fog onset phase was achieved. However, after onset the NWP-SCMs are of highly variable quality, but there appears to be a general trend for over development of fog, i.e. models produce fog which is too physically and optically thick, too quickly. There are some simple improvements (such as the inclusion of cloud droplet sedimentation) which can be applied to some models, but further improvements could require some significant parametrization development. This work has given some guidance as to where that work should be focussed, as we have shown that fundamental



parametrizations (such as cloud microphysics) are as uncertain and important in simulating fog development as implementing new feedback processes (such as aerosol interaction). However, there are still fundamental questions on the interaction between cloud, radiation and turbulence in fog which require further investigation.

Regarding forecasting applications, this work has shown that the early stages of fog development crucially impact its decay phase the following morning. This suggests that if real-time comparison of NWP forecast to observations can be conducted during the night-time, it could be used to help determine how accurate the NWP dissipation forecasts will be, allowing them to be manually adapted to give the best guidance to customers. Success has been seen with techniques like this in the past (Bergot, 2007), and with new and emerging observational platforms (such as UAVs), more detailed measurements of the fog properties could further improve customer guidance.

*Data availability.* The data is available from the authors upon request.

*Author contributions.* IB analysed the submitted results and wrote the manuscript. IB, WA, RB, AB, LD, RF, TG, EG, AH, AI, IK, JS and GS ran the model simulations. All authors contributed to the discussion, understanding and presentation of results, and preparation of the manuscript.

*Competing interests.* The authors declare that they have no conflict of interest.

*Acknowledgements.* WA thanks Greg Thompson of NCAR for help in understanding and setting parameters in the Thompson microphysics schemes in WRF. JS was supported by the Hans Ertel Centre for Weather Research of DWD (The Atmospheric Boundary Layer in Numerical Weather Prediction) grant number 4818DWDP4. RB was supported by MeteoSwiss (project number 123001738). This work used resources of the Deutsches Klimarechenzentrum (DKRZ) granted by its Scientific Steering Committee (WLA) under project ID bb1096. IK and SR acknowledge the Horizon 2020 Research and Innovation Programme (grant no. 821205).





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
