# Peer review of "Demistify: an LES and SCM intercomparison of radiation fog"

_Atmospheric Chemistry and Physics, 2021_

## Author Comment (AC1)

**Response to reviewer 1**

General comments:

The manuscript presents the results of an intercomparison of simulations of a radiation fog event from a wide variety of numerical models. Models are composed of Large Eddy Simulation (LES) codes and single column (SCM) versions of numerical weather prediction models. The event is an idealization of a case observed during the LANFEX experiment. Despite the great care in constraining models with the observed surface temperature, large differences are observed in fog properties produced by the different models.

First, it is with great enthusiasm that I learn that this is a first report on coordinated efforts focused on improving the numerical modeling of fog, directed at improving real-world applications of fog forecasts. The manuscript is well-written and provides great, albeit disconcerting, insights into the current state of fog numerical modeling of. A novel aspect of the study consists of comparing LES and SCM models on the same reference case. The manuscript shows in a compelling way the shortcoming of current modeling systems, including the LES where the reliance on the parameterization of turbulent mixing is not as strong as in operational numerical weather prediction models.

Below are some comments and suggestions to help clarify some points and hopefully further improve the manuscript.

Many thanks for the kind comments about our work!

Specific comments and recommended revisions:

Page 3, lines 47, 50-52: I am fine with the simplification of the event and agree that the strategy to force all models with the observed surface temperature allows for a more focused intercomparison. However, since comparison with real observations are presented later as the reference, you should justify that IOP1 did not involve other forcings (advection from drainage flows or other mesoscale circulations, upper tropospheric clouds etc). Please provide a brief description of these conditions and/or clearly point to the prior publication describing IOP1. That would help justify using real observations as the reference in your model intercomparison.

We have pointed the reader to the previous publication (Boutle et al 2018) which showed 3D NWP model simulations (i.e. all possible forcings) compared very well to an LES using the simplified setup. We also tested an SCM setup with advective forcings derived from the radiosonde profiles throughout the night, which showed little difference in fog evolution (below). This has been noted in the text.

[Figure]

Page 3, lines 49-50: Which temperature is used to constrain the models? Skin surface temperature? And how was it observed? Please clarify.

It was the surface skin temperature, measured with an infra-red radiation thermometer. This has been clarified, and a reference to Price et al (2018) for further details added.

Page 3, lines 69-70, here you could relate the parameters of the idealized simulation to the conditions at the Cardington site.

We've noted in the text that these parameters are chosen because they are representative of the observational site, and provided a reference to Price et al (2018) which gives some more discussion of the site setup.

Page 5, line 77: Models exhibiting positive fluxes are pointed out in the next section and the importance of behavior in fluxes in simulations are pointed out later in the paper, so this aspect seems more important than conveyed here. I would suggest revising the statement about the limited importance of fluxes (evapotranspiration).

We have revised this statement to clarify that the observed fluxes are negligible, and so the simplification should be of limited importance if the models can reproduce this behaviour.

Page 5, line 86: How do you define "reasonable"? Can you provide a more quantitative statement?

We really mean that they are close to, but just outside of the observational range, and have clarified this in the text.

Page 7, Figure 2: the curve showing the observed conditions does not appear clearly on the figure. I would suggest revising the figure so that the observations "pop out" more visually.

This was a deliberate choice. We wanted the focus of the plots to be on the model comparison, and the observations are shown less prominently to give some context and background. We agree they are a bit too faint on some plots though, so have thickened the lines to make them more visible.

Page 11, line 146: How does LW cooling lead to an erosion of LWP? LWC in the upper part of the fog layer should increase due to LW cooling. Please provide a more complete but brief explanation related to your statement.

This has been clarified in the text. I hope we are paraphrasing your paper correctly!

Page 11, 148, 152: Are the different behavior in the two versions of the Meso-NH model only due to the different aerosol loads, or more from differences in model formulation? Please explain more.

This is mainly a model formulation issue – the microphysical parametrizations are different. This has been clarified in the text.

Page 11, paragraph between lines 149 and 164: by surface heat flux, you are referring to the sensible heat flux correct? You should make that clear.

This has been corrected to clarify that it's the surface sensible heat flux being discussed.

Page 11, line 163: What is meant by "feed back" here? What interaction(s) are implied here? Please clarify.

We have clarified that the expected trend being discussed is for higher surface sensible heat fluxes to promote deeper fog layers, and RAMS does not produce this behaviour – it has a strong sensible heat flux but the shallowest fog layer of any LES.

Page 11, line 169, statement about better adaptation to the local environment: is it better or a deficiency in their BL turbulence scheme? Maybe UM does the mixing too quickly, but no mixing at all seems like a strange behavior under those conditions.

We have investigated this a bit further based on this comment. We believe the reasoning is the coarse vertical resolution used in the IFS model, with only 2 levels in the fog layer at this stage. The first layer is well-mixed, as you might expect given the surface flux. The second level, which is the one that appears most clearly on the figure, is dominated by the cloud top entrainment, and hence has a stable profile. We have clarified this in the text.

Page11, line 174: How was the latent heat flux in FV3 restricted? Where is this discussed? This point should be made clearer to the reader.

We've now explained this further in the text. A lower limit of zero was explicitly applied to the latent heat flux in the FV3-GFS model to enable fog to form.

Figures 5 and 6, same comment as for figure 2. The curves showing the observed values do not show clearly.

As above, we've made the observations line thicker to increase visibility.

Page 14, line 178, statement about unrestricted evaporation: I am not sure I understand this. How is it unrestricted? Why should it be restricted to be more realistic? Please clarify this statement.

We have clarified this in the text here, and when discussing the "unrestricted evaporation" in Section 2. What we mean is that the surface is treated like a sea-surface, i.e. it produces the maximum latent heat flux theoretically possible, rather than what a true land surface scheme would produce for a given soil moisture availability and stomatal resistance of the grass.

Page 14, line 180, statement about overlying cloud cover: This information should be provided earlier in the paper, likely with the discussion of Fig. 1 as the jump in LWP after 0800 UTC is a clear feature of the event that is not related to the behavior of the fog layer.

We've noted in section 2 that the real clearance was driven by the overlying cloud cover, and commented again in the discussion of Figure 1 that this is the reason for the large increase in LWP in the morning.

Page 15, lines 196-197: The point you are making about using real time observations to evaluate model performance is really interesting. It could be useful here to expand/speculate about the suite of observations required to do this.

We have added some text on this to the paper as suggested. We think the observations are many of those analysied in the paper, i.e. radiosonde profiles, LWP measurement, surface fluxes, as these will allow an assessment of whether the model is over- or under-developing fog, and therefore whether it is likely to dissipate earlier or later in the morning than forecast.

Page 15, Figure 9: I do not see any dashed lines in the SCM plot. Does that mean that aerosol loads have no impact in the dissipation phase of the fog for all of the models? Despite the differences in overnight fog evolution? This is strange and should be discussed.

This is correct. We have now noted in the text that the dissipation is closely tied to individual models' behaviour, and not the general characteristics of the setup or fog development. A more focussed intercomparison on the dissipation phase is probably required to make progress here.

Page 15, line 198, statement about fog burning-off versus lifting: I thinks this has a lot to do with fog depth, strength of fog-top inversion and entrainment, and possibly advections of temperature and moisture. You could expand on your statement a little more for greater clarity.

We have added a comment on the mechanisms which determine the fog dissipation or lifting, and whether or not they are likely to be included in these results.

Page 16, line 207: How is visibility calculated in the different models? Is visibility estimated at the same level for all models? What about the models where the lowest level is much higher than 2 m? Also is the same visibility parameterization used? What are the differences in the relationship uses among the various models? Please clarify these points.

We have listed the visibility parametrization used in each model in Table 4, and added some discussion to the text. The parametrizations are model dependent, and show a mix of empirical or physical techniques. The height applied is does vary between the models – for most it is in the screen level (1.5-3m) range, but for some (shown in Tab 4), the lowest model is outside of this range and used directly. Other models which don't have a vertical level at this height, interpolate the input variables for the visibility parametrization to this level first.

Page 16, lines 212-213: I am not 100% convinced about explaining the similar behavior by constraining to a common temperature at the surface. Could you provide further evidence to justify the statement? The large differences in observed in LWP and water deposition rates at the surface would suggest significant differences in screen-height LWC, the main variable in determining visibility. I think the reader would like more evidence here.

We've included an additional figure showing the screen level temperature produced by each of the models we present the visibility from, demonstrating that they are all in very close agreement. Sadly we did not request a screen level LWC diagnostic, so we cannot present that. However, given the differences in the visibility calculations discussed above, it's unclear whether these being similar (or not) would support the point we're trying to make, which is that the strong agreement between visibility forecasts does not tell us much about the fog representation within the models.

Page 18, line 228: the focus of this section is only on microphysics (and rightfully so!), but the title suggests that a wider range of parameterizations are considered. I would suggest adding "Microphysics" at the beginning of the section title.

This has been changed as suggested.

Page 20, line 285: I agree that improvements have been made since Bergot's intercomparison. However, the sample of cases on which models have been tested remains very small. I think this is a point that should be underlined here.

We have added a sentence to the end of this paragraph, noting that this intercomparison just considered a single case, and more should certainly be done in different locations and with different forcings.

Page 20, lines 288-289: We should be careful not to suggest the importance of droplet sedimentation as a recent discovery. This has been known for a long time (prior studies by Brown and Roach in 1976, Bergot and Guédalia in 1994 etc.). I think the current lack of inclusion in some of

the models rather speak to how the problem of fog modeling has been neglected by the model developers in some of the research and operational centers. Perhaps a statement to that effect would be appropriate? Not absolutely necessary though. But I hope your study will contribute at changing that. Well done.

We agree, and certainly our intent for this sentence was to point out that we feel it's bad that many models are still missing this process, the importance of which has been known for many years. Although we also acknowledge that model development is a tricky process with lots of competing interests, and so don't want to be too hard on those models which are lacking the parametrization. We have changed "could" to "should" which hopefully makes a more forceful recommendation to modellers.

Page 21, lines 298-299: Again, perhaps some suggestions of the needed observations would be helpful?

We've not repeated the list provided earlier for brevity, but have added droplet spectra as an example of something that UAVs could provide in real-time.

Great work again! I would love to see more studies like this.

---

## Author Comment (AC2)

**Response to reviewer 2**

General Comments

This paper describes the initial phase of a model intercomparison project focused on the modelling of fog. Focussing primarily on the atmospheric development of fog, 5 large eddy simulation (LES) and 10 single-column models (SCM) are run with prescribed high and low aerosol/CDNC in the simulations while surface properties were constrained following a slightly idealised version of the LANFEX IOP1. They found that there were large differences between the models and highlighted the importance of the inclusion of processes such as cloud droplet sedimentation as well as the sensitivity of parameterisations within the microphysics.

It is great to see the international community come together to tackle the problem of fog modelling. This manuscript provides a good overview of the capabilities and limitations of the current state of the art models for fog. The comparison of the models for the same case and with similar constraints is useful. I think this paper provides a solid baseline for further studies and would recommend it to be accepted for publishing, with a few minor revisions.

Many thanks for the positive assessment of our work.

Specific comments and technical suggestions

Figures: The observations in all figures except 1 ,11 & 12 are very hard to see. Maybe plot it last so that it lies on top of the simulations, and/or use a darker shade in your colouring.

This was a deliberate choice. We wanted the focus of the plots to be on the model comparison, and the observations are shown less prominently to give some context and background. We agree they are a bit too faint on some plots though, so have thickened the lines to make them more visible.

Line 6: "under high aerosol or cloud droplet number concentration (CDNC) conditions."

This has been changed.

Line 30: Maybe say "The current intercomparison" instead of "This intercomparison"? As you were just talking about the previous one it was not immediately clear to which one "this" refers to.

This has been changed as suggested

Line 47: How exactly did you "idealise" IOP1? It is not clear.

This is discussed in the paragraph below this sentence, which has been expanded to give more details on the case setup and idealisation thereof as suggested. The main simplification is that no forcing other than surface temperature is applied, i.e. no forcing of the horizontal winds, and no overlying cloud cover is advected over the site.

Line 82-84: It might be useful to briefly describe the relationship between LWP and fog/cloud for readers less familiar with the phenomena. i.e. how do you distinguish between fog and cloud when looking just at LWP? Assuming the observed LWP towards the end of the period is cloud, do you expect the models to simulate that too?

Some text describing this has been added as suggested. For the bulk of the simulation, there is only fog, hence this is what the LWP is showing. We've noted that the presence of the cloud at the end shows up in the observed LWP, and is not simulated and therefore shouldn't be reproduced by the models.

Line 165: To which figure are you referring to here?

We've added a pointer to Figs 5c and 6b here.

Table 4: It would be nice to have a reminder of which models are SCM and which are LES in this table. Either as an extra column or in brackets after the model for the ones that doesn't have it in the name.

This has been added as suggested.

Line 225-227: Yet the fog seems to persist for too long into the daytime for the LES models as well?

We've added more discussion earlier in this section about how the real dissipation is not represented by the intercomparison participants, and therefore we make no attempt to compare either the LES or SCMs to the real dissipation. This sentence was trying to link together the fact that fog which is too thick will, in general, likely be difficult to dissipate, and therefore persist for too long. We have clarified the sentence slightly.

Line 254-255: To me it doesn't look like the microphysical parameterisation has a larger effect than the prescribed CDNC values. c10,u=10 have a similar LWP to c50,u=0, but all the other c50 simulations are well above the rest, whereas the c10 simulations only have a slight increase with increasing u between 2.5 and 10. So except for setting u=0, you'd need larger CDNC for u to have more effect. But your point stands that it does make a difference.

We've modified this to simply state that the mu value can have similar sized effects to the prescribed CDNC.